# A reductionist paradigm for high-throughput behavioural fingerprinting in *Drosophila melanogaster*

**Hannah Jones[1], Jenny A Willis[2], Lucy C Firth[2], Carlo NG Giachello[2], Giorgio F Gilestro[1]***

[1]Department of Life Sciences, Imperial College London, London, United Kingdom; [2]Syngenta, Jealott's Hill International Research Centre, Bracknell, United Kingdom

**Abstract** Understanding how the brain encodes behaviour is the ultimate goal of neuroscience and the ability to objectively and reproducibly describe and quantify behaviour is a necessary milestone on this path. Recent technological progresses in machine learning and computational power have boosted the development and adoption of systems leveraging on high-resolution video recording to track an animal pose and describe behaviour in all four dimensions. However, the high temporal and spatial resolution that these systems offer must come as a compromise with their throughput and accessibility. Here, we describe *coccinella*, an open-source reductionist framework combining high-throughput analysis of behaviour using real-time tracking on a distributed mesh of microcomputers (ethoscopes) with resource-lean statistical learning (HCTSA/Catch22). Coccinella is a reductionist system, yet outperforms state-of-the-art alternatives when exploring the pharmacobehaviour in *Drosophila melanogaster*.

*For correspondence: giorgio@gilest.ro

Competing interest: The authors declare that no competing interests exist.

## eLife assessment

This study presents an **important** open-source resource for high-throughput behavioral screening. The protocols employ inexpensive, off the shelf hardware, and allow real-time analysis of hundreds of behaving flies. Although these protocols were developed using *Drosophila melanogaster*, they could easily be applied to other models. The evidence in support of the conclusions is **solid** and the revisions carried out by the authors go a long way towards providing the user with an integrated system that is also more user-friendly.

## Introduction

The nervous system integrates stimuli, internal states, expectations, and previous experience to regulate behavioural output. Describing, quantifying, and modulating behaviour are critical aspects of modern neuroscience and, ever since its inception, the field has spent considerable effort into building and sharing paradigms or tools aimed at objectively and reproducibly quantify behaviours in the most disparate animal models, to the point that this exercise is now recognised as an exciting subfield of neuroscience in its own right: ethomics (**Brown and de Bivort, 2018**; **Datta et al., 2019**). As the portmanteau name itself suggests, ethomics is not just about describing behaviour (*etho*-) but also about doing so in a high-throughput fashion (-*omics*), collecting data simultaneously from a large number of individuals, which can remain undisturbed throughout recording. Irrespective of the behaviour or the animal model to be analysed, the first compromise a researcher will face when choosing a tool for behavioural quantification will always be between throughput and resolution: a high-throughput analysis will allow for powerful experimental manipulations – such as genetics or

pharmacological screens – offering unbiased approaches in identifying neuronal circuits, genes, molecules underpinning behaviour; high-resolution analysis, on the other hand, promises to identify and discriminate even minuscule differences that may not be immediately visible to the human eye, and to label behaviours into identifiable classes (e.g. 'grooming', 'courting', and 'shaking') that may be more relevant to researchers interested in modelling disease or in anthropomorphic descriptions. In the past years, the field has generally converged towards the adoption of high-resolution video recording of activity, in some cases adopting cameras that have milliseconds temporal resolution or developing setups that provide depth information for three-dimensional reconstruction of motion or posture (*Wiltschko et al., 2015*; *Hsu and Yttri, 2021*; *Nath et al., 2019*; *Pereira et al., 2019*; *Gosztolai et al., 2021*). Given the recent evolution in machine learning and progresses in computational power, even these high-resolution analyses can be at least in part compatible with high-throughput approaches (*Wiltschko et al., 2020*), especially when employed on small invertebrate animal models (*McDermott-Rouse et al., 2021*; *Ayroles et al., 2015*; *Branson et al., 2009*; *Kabra et al., 2013*) or when aided by robotic handling (*Alisch et al., 2018*). These systems, however, can still be prohibitively expensive for most laboratories, and not easily compatible with throughput in the '*omics* scale. Moreover, besides the technical urge of removing entry barrier and make ethomics an accessible tool, an equally important underlying question concerns what is the minimal amount of information that needs to be extracted to identify and classify behaviour. Do we always necessarily gain information from extracting micro-postural features or by analysing activity in three dimensions? To what extent this may actually add counterproductive biological noise to some assays?

Here, we introduce *coccinella*: a new experimental framework that combines high-throughput, inexpensive, real-time ethomics (*Geissmann et al., 2017*) with state-of-the-art statistical analysis (*Fulcher and Jones, 2017*; *Lubba et al., 2019*) to characterise and discriminate complex behaviours using a reductionist approach based solely on one simple feature (https://lab.gilest.ro/coccinella). *Coccinella* builds on ethoscopes (*Geissmann et al., 2017*), an accessible open-source platform, to extract, in real-time, activity information from flies. Despite its minimalist nature, *coccinella* outperforms state-of-the-art alternatives in recognising the pharmacobehavioural space, providing better discernibility at a fraction of the cost, thus opening a new path to high-throughput ethomics.

## Results

*Drosophila* ethomics studies generally rely on image acquisition through so-called industrial cameras, able to collect videos with high temporal and spatial resolution and featuring mounts for a large selection of lenses. Some of the cameras commonly used for these purposes (e.g. FLIR, Point Grey, Basler) (*Mathis et al., 2018*) are expensive and normally employed in close-up imaging that microscopically highlights the smallest anatomical features of the animal but at the same time greatly limits the number of experimental subjects that can be recorded by a single device. Normally, one or few more camera would be connected to a dedicated powerful computer for acquisition and storage of videos. The cost and physical footprint of these setups makes them incompatible, at least for most laboratories, with high-throughput simultaneous acquisition. To lower this barrier, we created a framework that employs the distributing computing power of ethoscopes (*Geissmann et al., 2017*), thus allowing for inexpensive analysis of activity in hundreds or thousands of flies at once. Ethoscopes are open source and can be manufactured by a skilled end-user at a cost of about £75 per machine, mostly building on two off-the-shelf component: a Raspberry Pi microcomputer and a Raspberry Pi NoIR camera overlooking a bespoke 3D-printed arena hosting freely moving flies. The temporal and spatial resolution of the collected images depends on the working modality the user chooses. When operating in offline mode, ethoscopes are capable to acquire 720p videos at 60 fps, which is a convenient option with fast moving animals. In this study, we instead opted for the default ethoscope working settings, providing online tracking and real-time parametric extraction, meaning that images are analysed by each Raspberry Pi at the very moment they are acquired (*Figure 1b*). This latter modality limits the temporal resolution of information being processed (one frame every 444 ± 127 ms, equivalent to 2.2 fps on a Raspberry Pi3 at a resolution of 1280 × 960 pixels with each animal being constricted in an ellipse measuring 25.8 ± 1.4 × 9.85 ± 1.4 pixels – *Figure 1a*) but provides the most affordable and high-throughput solution, dispensing the researcher from organising video storage or asynchronous video processing for tracking the animals. In the work here described, flies moved freely in a circular two-dimensional space with a diameter of 11.5 mm designed to maintain the animal in the walking

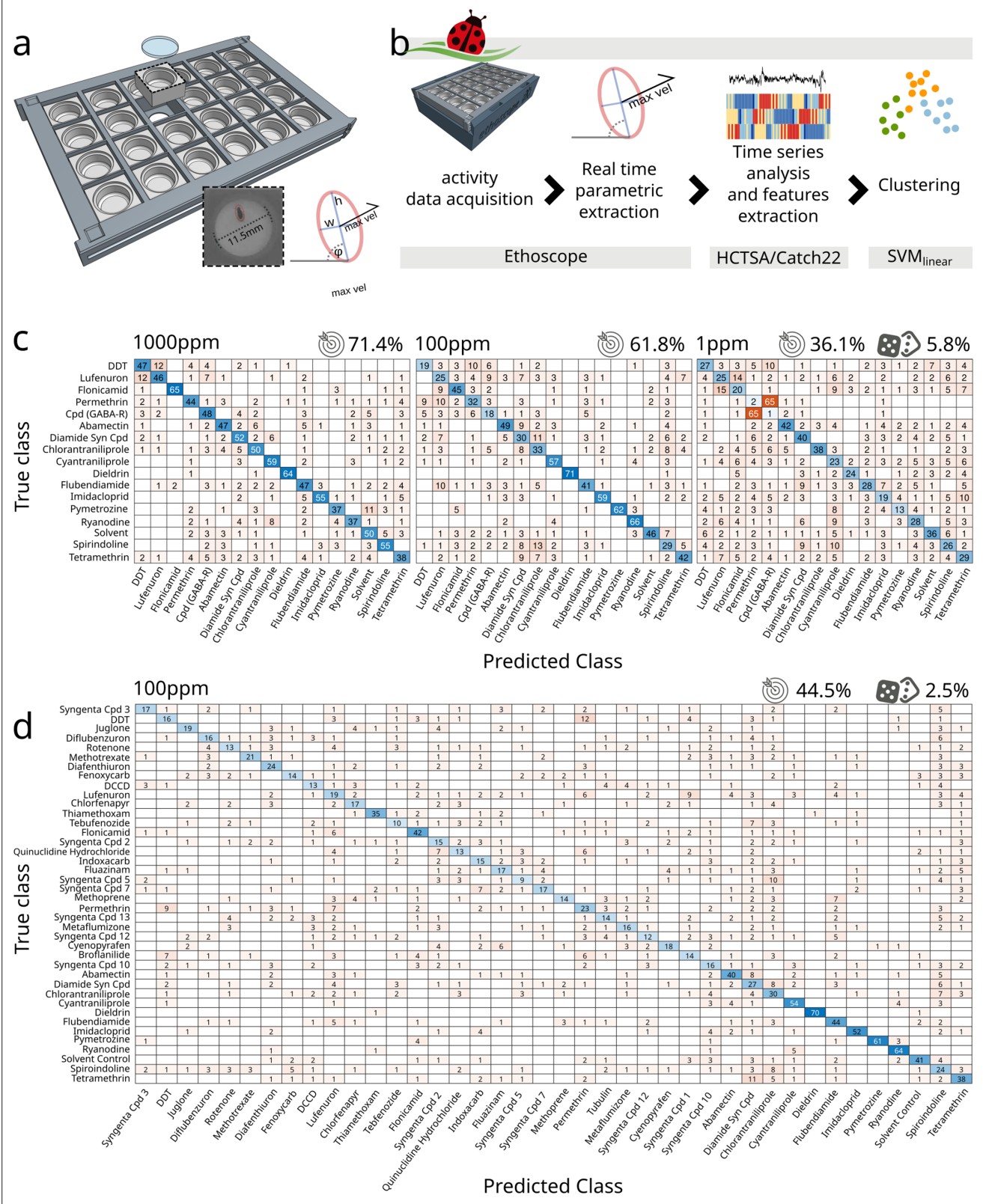

**Figure 1.** *Coccinella* successfully classifies pharmacobehaviours. (**a**) Bespoke 3D-printed arena can house 24 individual flies in a two-dimensional circular space. Each arena measures 11.5 mm in diameter and allows for back illumination either by visible or infrared light sources embedded in the ethoscope base. The red ellipse shows the data being extracted by the ethoscope in real time. *w, h*: width and height of the ellipse inscribing the animal. *φ*: the angle of the ellipse in reference to the region of interest. *max. vel.*: maximal velocity over the last 10 s. (**b**) Flowchart describing the analytical

*Figure 1 continued on next page*

Figure 1 continued

pipeline and the tools that compose *coccinella*. (**c**) Confusion matrices for treatments with 16 compounds and a solvent control, with drugs used at concentrations of 1000 ppm (left), 100 ppm (centre), and 1 ppm (right). The target icon indicates calculated accuracy while the rolling die indicate the accuracy of a random classifier. (**d**) Confusion matrix for the largest panel of 40 treatments at 100 ppm.

The online version of this article includes the following figure supplement(s) for figure 1:

**Figure supplement 1.** Successful classification of a smaller group of 11 compounds.

**Figure supplement 2.** Effect of drug resistance conferring mutations on Coccinella's performance.

position (**Simon and Dickinson, 2010**) while venturing on solidified agar providing nutrients alone or nutrients and drugs. In previous analysis of activity and sleep (**Geissmann et al., 2017**; **Geissmann, 2018**), we found that the maximal velocity of the fly over a period of 10 s best described the basic motion features of the animal, allowing us to accurately differentiate between different activity patterns, such as walking, grooming, and feeding (**Geissmann et al., 2017**). We therefore adopted this measure for *coccinella* too, ultimately producing monodimensional time series of a behavioural correlate, which were then digested using highly comparative time-series analysis (HCTSA) (**Fulcher and Jones, 2017**), a computational framework that effortlessly subjects time series to more than 7700 literature-relevant statistical tests, looking for meaningful discriminative features. Features successfully extracted through HCTSA were then used to classify behaviour using a linear support vector machine ($SVM_{linear}$) (**Chatterjee et al., 2022**; **Figure 1b**) and here presented and compared using confusion matrices (**Figure 1—figure supplement 1a**). To test the ability of this system to discriminate behaviour, we started by exploring the pharmacobehavioural space of flies fed with a panel of known or putative neurotropic chemicals, comprising molecules previously described in the literature along with uncharacterised ones being considered as potential insecticides (**Figure 1c, d** and **Supplementary file 1**). Using an initial panel of 17 treatments (16 drugs and 1 solvent control) we were able to discern compounds with an accuracy of 71.4% (vs. 5.8% of a random classifier – **Figure 1c**). Some compounds induced behaviours with a particularly high predictive fingerprint: dieldrin, for instance, was predicted with an accuracy of 94% and flonicamid with an accuracy of 87%. For others, our system fared more poorly (e.g. tetramethrin showed 41% accuracy). In all cases, however, the relative confusion was negligible, with all compounds being correctly identified as the first choice and with the first predicted compounds having, on average, a score that was 15 times greater compared to the second-best choices (**Figure 1c** – min.: 3.3×; max.: 65×). To validate our framework and exclude artefacts operated by overfitting biologically irrelevant information we followed two lines of control. Firstly, we fed flies with lower concentrations of the same compounds (**Figure 1c**). Feeding flies with different concentrations of drugs unsurprisingly showed a different effect on short-term lethality (**Figure 1—figure supplement 1b**), with several compounds hitting a 25% lethality rate before the end of the experiment when fed at the highest concentration (1000 ppm – **Figure 1—figure supplement 1b**). Lowering the compound concentration, the predictive accuracy of the system decreased from 71.4% (1000 ppm) to 61.8% (100 ppm), falling to 36.1% with the lowest concentrations (1 ppm), indicating that the system does operate on pharmacologically induced, biologically meaningful behavioural correlates (**Figure 1c**). A similar drop in accuracy was observed using a smaller panel of 12 treatments (**Figure 1—figure supplement 1c**). As a second line of work to test specificity, we obtained genetic mutants known to be resistant to specific pharmacological treatments: the $para^{L1029F}$ allele encodes for a version of the α-subunit of voltage-gated sodium channel conferring resistance to dichlorodiphenyl-trichloroethane (DDT) and pyrethroids (**Kaduskar et al., 2022**); the $Rdl^{A301S}$ allele encodes for a version of the ligand-gated chloride channel conferring resistance to dieldrin and fiproles (**Remnant et al., 2014**). Challenging these mutants with their respective compounds created confusion in the clustering algorithm for which discerning between drug treatment and solvent control became a harder task, especially in the case of DDT and $para^{L1029F}$ (**Figure 1—figure supplement 2a**). The observed drop in accuracy suggests again that *coccinella* is working on biologically relevant behavioural signatures and the fact that some discrimination can still be observed with targets harbouring point mutations – which should severely affect compound efficacy – is indicative of high sensitivity.

Having established the accuracy and sensitivity of the system, we next wanted to test its usefulness in a genuine high-throughput scenario. We subjected a total of 2192 flies to a panel of 40 treatments (**Figure 1d**), mostly featuring known compounds but also two unexplored molecules (**Supplementary**

*file 1*). Given that the 100 ppm intermediate concentrations showed the best compromise between accuracy and lethality in the previous pilot experiment (*Figure 1—figure supplement 1b*), we performed this larger screen using compounds diluted at 100 ppm only. Even with such a large panel, the system was able to first-guess 39 out of 40 of the tested treatments (the only exception being the Syngenta Compound #5) with an overall accuracy of 44.5% vs. 2.5% of the random classifier.

A reductionist approach served us well so far, identifying with remarkable accuracy even subtle changes when we explored the pharmacobehavioural space of a large number of neuroactive compounds in wild-type and mutant flies. But how does it compare to other more established paradigms? *Coccinella* is arguably to be preferred in terms of accessibility and throughput, but what is the amount of useful information that we are sacrificing by adopting a reductionist approach? To quantify any possible loss in information content, we ran a series of parallel experiments in which flies were fed with a selected panel of 12 treatments (11 drugs and a solvent control, *Figure 2*) and their behaviour analysed either using *coccinella* or using other widely adopted state-of-the-art methods, which started with high-resolution imaging and employed supervised machine learning for pose-estimation Deep-LabCut; *Mathis et al., 2018* followed by unsupervised identification of behavioural grammar (B-SOiD; *Hsu and Yttri, 2021*). To widen the range of comparisons, data were then either immediately clustered using different common clustering algorithms (*K*-nearest neighbours, random forest) or first processed through a smaller, selected subsample of the HCTSA array (Catch22; *Lubba et al., 2019*) before being clustered using SVM$_{linear}$ (*Figure 2a*). In this challenge, *coccinella* unambiguously identified 10 out of 12 compounds, with poor performance only for two of them (flubendiamide and tetramethrin) and an overall accuracy of 42.4% vs. 8.3% of the random classifier (*Figure 2b*). Surprisingly, none of the state-of-the-art high-resolution paths did better than this. The combination of pose-estimation → grammar extraction → random forest classification scored as the second best, with an accuracy of 25.4% but with only three compounds being unambiguously identified (*Figure 2c*). The same experimental dataset clustered with even poorer performance when using *K*-nearest neighbours (*Figure 2e*) and even the application of HCTSA features extraction to the B-SoID output still could not match the accuracy observed with *coccinella*'s reductionist approach (*Figure 2d*). This analysis is not meant to be conclusive. We expect that some alternative combination of state-of-the-art approaches will probably manage to match or likely improve over *coccinella*'s performance, yet the fact we could obtain such an impressive result with a system that is arguably unmatched in terms of throughput and economic cost is, alone, an argument that gives new weight to this (and future) reductionist approaches.

Finally, to push the system to its limit, we asked *coccinella* to find qualitative differences not in pharmacologically induced changes in activity, but in a type of spontaneous behaviour mostly characterised by lack of movement: sleep. In particular, we wondered whether *coccinella* could provide biological insights comparing conditions of sleep rebound observed after different regimes of sleep deprivation. *Drosophila melanogaster* is known to show a strong, conserved homeostatic regulation of sleep that forces flies to recover at least in part lost sleep, for instance after a night of forceful sleep deprivation (*Shaw et al., 2000*; *Hendricks et al., 2000*). We previously showed that the extent of sleep rebound observed after sleep deprivation only loosely correlates with the amount of lost sleep (*Geissmann et al., 2019a*) and it remains an open question whether similar amounts of sleep rebound may in fact differ from each other in some inscrutable feature that would underpin a different 'sleep depth' (*Wiggin et al., 2020*; *French et al., 2021*), similar to what it is believed to happen in mammals. Here, we analysed a dataset of 727 flies that experienced different regimes of mechanically enforced sleep deprivation during the 12 hr of the night (*Figure 3*). Flies were housed in tubes that would rotate after a set time of inactivity ranging from 20 to 1000 s leading to different degrees of sleep restriction (*Figure 3a*, dataset from *Geissmann et al., 2019a*). In this experimental paradigm, all treatments led to a statistically significant rebound compared to the undisturbed control animals (*Figure 3b*). We then ran *coccinella* on the two subsets of the panel: the baseline data, acquired the morning before the sleep deprivation (*Figure 3c*), and the rebound data, on the morning after (*Figure 3d*). Unsurprisingly, we could not detect any internal biological difference in the pre sleep deprivation control set, featuring flies of identical genotype and age housed in different tubes before the sleep deprivation treatment. In these conditions, *coccinella* could not discern, and performed exactly as a random classifier would (9% vs. 9% – *Figure 3c*). However, analysis of those same animals during rebound after sleep deprivation showed a clear clustering, segregating the samples in two subsets with separation around the 300 s inactivity trigger (*Figure 3d*). This result is important for two reasons: on one hand,

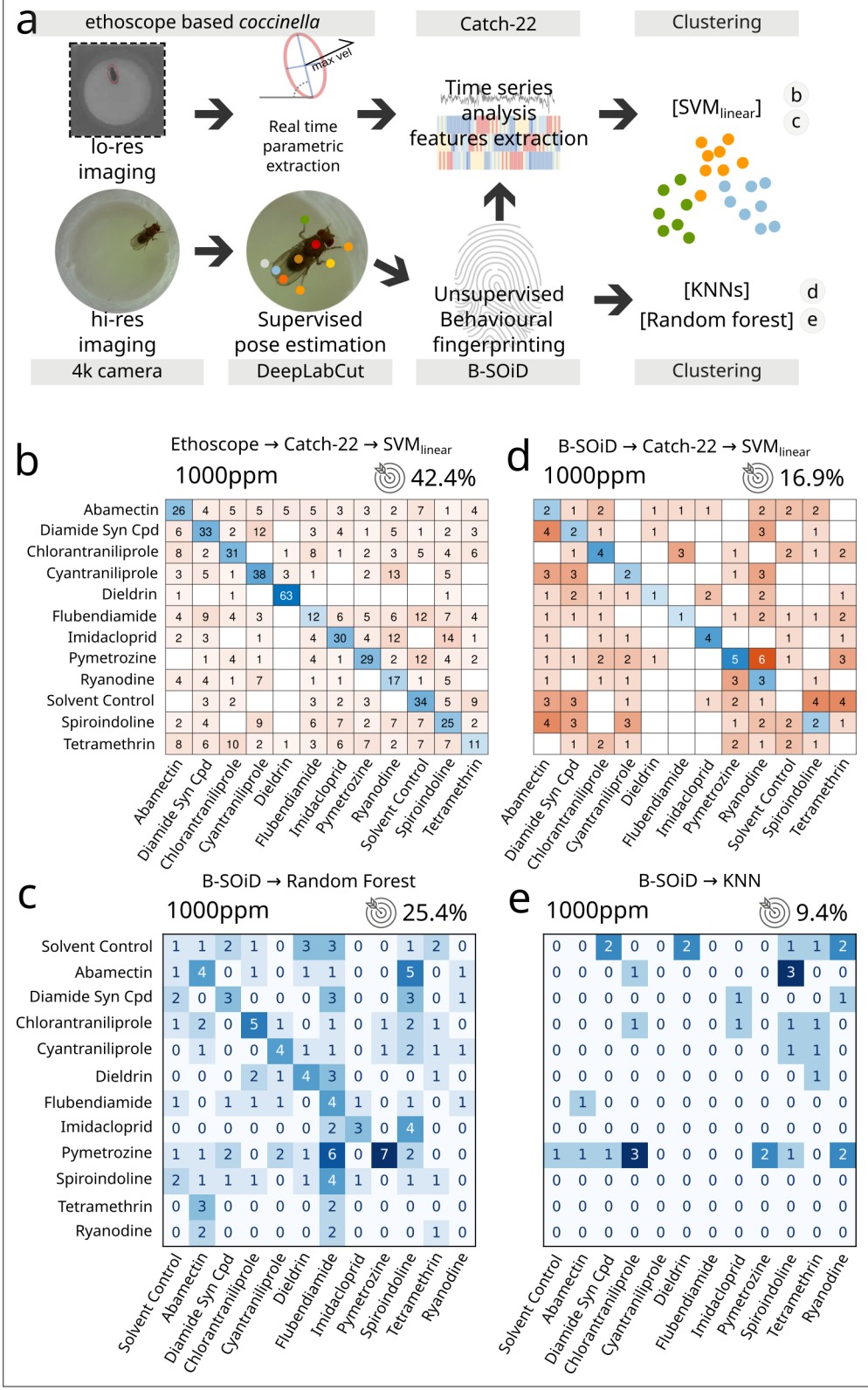

**Figure 2.** Comparison between *coccinella* and the state-of-the-art. (**a**) Experimental pipeline illustrating the four experimental analyses. (**b**) Confusion matrix for 12 treatments (11 drugs at 1000 ppm and 1 solvent control) analysed using *coccinella*. (**c**) Same experimental treatments as in b, analysed using the DeepLabCut → B-SoID → random forest pipeline starting from high-resolution images. The random forest classifier was trained on a 4:1

*Figure 2 continued on next page*

*Figure 2 continued*

training:testing ratio. (**d**) Same as c but with Catch22 identification and support vector machine (SVM) clustering after B-SoID grammar dissection. This is a hybrid treatment combining highly comparative time-series analysis (HCTSA) feature extraction to the high-resolution pipeline. (**e**) Same as c but using *K*-nearest neighbours (KNN) as cluster algorithm. KNN required a much higher training:testing ratio of 9:1, dramatically reducing the size of the testing dataset. The accuracy of a random classifier for all matrices on this figure would be 8.3% (not shown on figures for lack of space).

it provides, for the third time, strong evidence that the system is not simply overfitting data of nought biological significance, given that it could not perform any better than a random classifier on the baseline control. On the other hand, *coccinella* could find biologically relevant differences on rebound data after different regimes of sleep deprivation. Interestingly enough, the 300 s threshold that *coccinella*

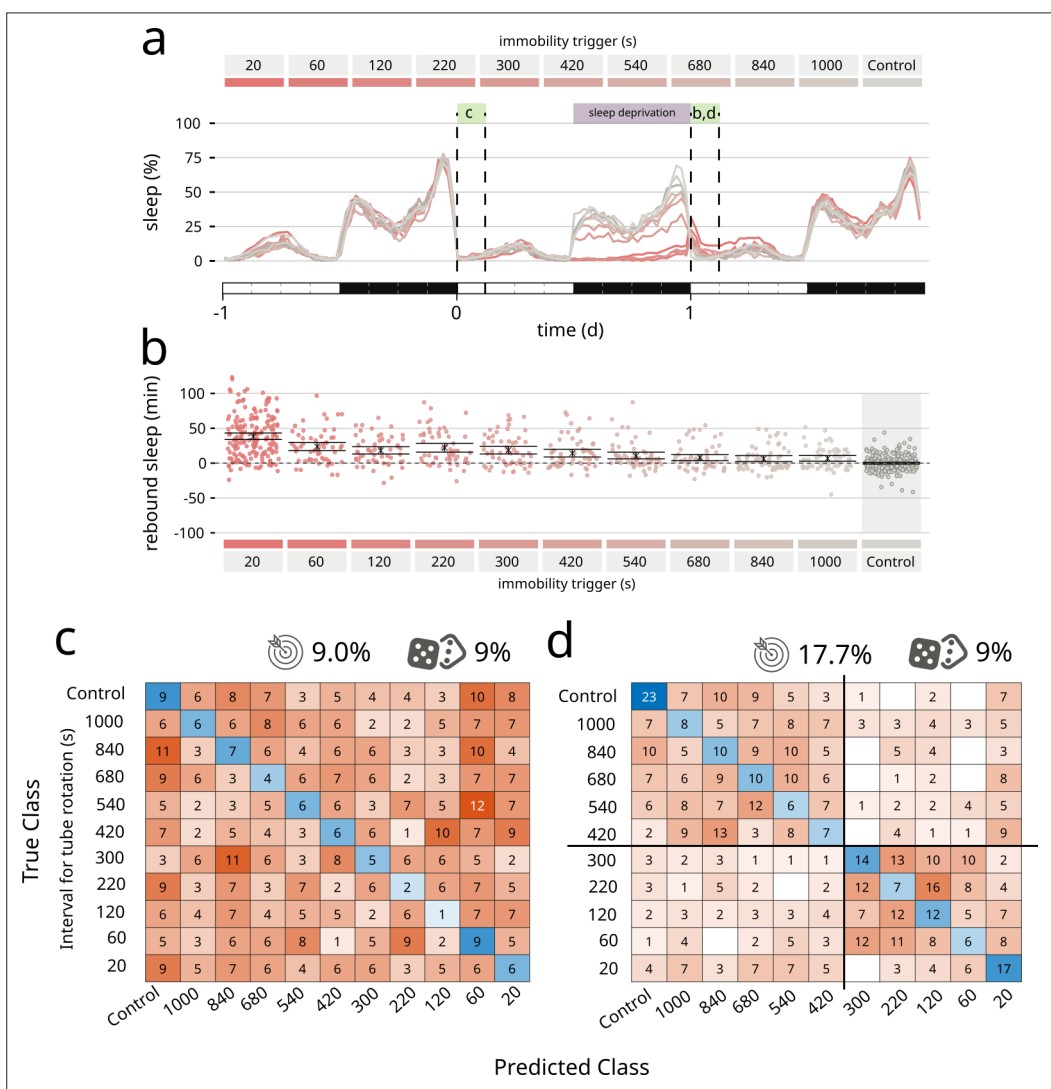

**Figure 3.** *Coccinella* finds differences in type of sleep rebound after sleep deprivation. (**a**) Sleep profile of flies over the period of 3 days. A 12-hr sleep deprivation regime starts at the beginning of the dark phase of day 0 (purple bar). The 3-hr windows labelled with green boxes were analysed by *coccinella* in search of meaningful differences. The letters above refer to the panels using data in those time windows. (**b**) Extent of rebound as observed following sleep deprivation as performed in a. Panels a and b reproduce data from *Geissmann et al., 2019a*. (**c**) Confusion matrix showing the classification using *coccinella* of the baseline time series. No accuracy gain compared to the random classifier. (**d**) Confusion matrix of the rebound data. The classification finds two clusters, separated by the 300 s threshold (thick black lines).

independently identified has a deep intrinsic significance for the field, for it is considered to be the threshold beyond which flies lose arousal response to external stimuli, defining a 'sleep quantum' (i.e. the minimum amount of time required for transforming inactivity bouts into sleep bouts; *Shaw et al., 2000*; *Hendricks et al., 2000*; *Joyce et al., 2023*). Coccinella's analysis ran agnostic of the arbitrary 5 min threshold and yet identified the same value as the one able to segregate the two clusters, thus providing an independent confirmation of the 5-min rule in *D. melanogaster*.

## Discussion

System neuroscience is living a period of renaissance, and *Drosophila* is driving this revolution strong of the first full-brain connectome, a plethora of new genetic reagents that allow thermo- and opto-genetic manipulations, a galore of genetic transformants for circuit tracking and manipulation, and multiple tools for large-scale quantification of behaviour. Progresses in machine learning and computer power have had a massive impact on the field of ethomics, especially in achieving levels of anatomical tracking that allow mapping of the tiniest movements on an experimental animal model with the highest temporal resolution and with little human supervision (*Pereira et al., 2019*; *Mathis et al., 2018*). Most of these systems, however, rely on relatively expensive setups and do not scale easily to high-throughput experimental paradigms. They are ideal – and irreplaceable – to identify behavioural patterns and study fine motor control but may be undue for other uses. Here, we introduce a new framework, *coccinella*, that merges an open-source, economically accessible hardware platform (etho-scopes; *Geissmann et al., 2017*; *Geissmann et al., 2019b*) with a powerful toolbox for statistical analysis and clustering (HCTSA, *Fulcher and Jones, 2017*/Catch22, *Lubba et al., 2019*). Coccinella is a reductionist tool, not meant to replace the behavioural categorisation that other tools can offer but to complement it. It relies on Raspberry PIs as main acquisition devices, with associated advantages and limitations. Ethoscopes are inexpensive and versatile but are limited in terms of computing power and acquisition rates. Their online acquisition speed is fast enough to successfully capture the motor activity of different species of *Drosophilae* (*Joyce et al., 2023*), but may not be sufficient for other animals moving more swiftly, such as zebrafish larvae. Moreover, *coccinella* cannot – and is not meant to – apply labels to behaviour ('courting', 'lounging', 'sipping', 'jumping', etc.) but it can successfully identify large behavioural phenotypes and generate unbiased hypothesis on how behaviour, and a nervous system at large, can be influenced by chemicals, genetics, artificial manipulations in general. Here, we provided evidence that *coccinella* can be used to successfully explore and compartmentalise the pharmacobehavioural space and also showed that a reductionist approach can be employed to discern otherwise invisible shades of a very subtle naturally occurring behaviour: sleep. The success of *Drosophila* as experimental model was built on the many genetic screens of the 1900s. We propose *coccinella* as an accessible, pivotal tool to boost again this important line of work in any laboratory, without funding or access to technology being a discriminative factor.

## Methods
### *Drosophila* rearing
Fly lines were maintained on a 12-hr light:12-hr dark (LD) cycle and raised on polenta and yeast-based fly media (agar 96 g, polenta 240 g, fructose 960 g, and Brewer's yeast 1200 g in 12 l of water). Canton-Special (CS) *D. melanogaster* were used as the wild-type line for all experiments.

### Drug-resistant mutants
$Rdl^{A301S}$ is derived from $Rdl^{MDRR}$ (RRID:BDSC_1675), an *Rdl* allele isolated from a natural population in Maryland (*Ffrench-Constant et al., 1990*), and underwent isogenisation and selection on dieldrin to eliminate the metabolic resistance and maintain the dieldrin target site resistance (*Blythe et al., 2022*). The $Rdl^{MDRR}$ was obtained from the Bloomington *Drosophila* Stock Center. For *para*: the L1029F mutation, located in the voltage-gated sodium channel paralytic, has been extensively reported to confer resistance to DDT and pyrethroids in many other insect species (called kdr, knockdown resistance, reviewed in *Arena et al., 1992*). In the *Drosophila* gene, kdr maps to L1029F and is equivalent to the often cited L1014F in other insects (e.g. *Musca domestica*; *Dong, 2007*). The kdr L1029F mutation in *Drosophila* Para was introduced via CRISPR/Cas9-mediated genome editing (see below).

This genome edited generated mutation resulted in a similar resistance to DDT as previously reported (*Samantsidis et al., 2020*).

## Generation of Para$^{L1029F}$ via CRISPR-CAS9-based genome editing

CRISPR/Cas9-mediated genome editing was used to introduce a point mutation L1029F in para-PBG isoform, CTT to TTT, L to F by homology-dependent repair using one guide RNA and a dsDNA plasmid donor. The strategy design, molecular biology, and screening were completed by Well Genetics Inc, Taiwan (R.O.C.). The cassette PBacDsRed contains Piggy Bac 3′ terminal repeats, the selection marker 3xP3-DsRed, and Piggy Bac 5′ terminal repeats. The selection marker 3xP3-DsRed contains Piggy Bac 3′ terminal repeats, 3x Pax3 and hsp70 promoter, DsRed2, SV40 3′UTR, and Piggy Bac 5′ terminal repeats. The dsRed marker facilitates the genetic screening and was excised by Piggy Bac transposase. Only one TTAA motif was left after transposition embedded in mutated intron sequence, and create a mutation G to A on X:16,486,649; X:16,486,649–X:16,486,646, CTAA to T TAA in intron. The CRISPR Target Site [PAM]: CACAAGATTGCCGATGACAA[CGG]. Guide RNA Primers: Sense oligo5′-CTTCGCACAAGATTGCCGATGACAA and Antisense oligo5′-AAACTTGTCATCGGCA ATCTTGTGC. Upstream Homology Arm: 1027 bp, the +34,097 nt to +35,123 nt from ATG of para. Forward Oligo5′-GTTCACCAAACTCGGAATCG; Reverse Oligo5′-GTGGCCAAGAAGAAGGGAAT . Downstream Homology Arm: 1022 bp, the +35,128 nt to +36,149 nt from ATG of para Forward Oligo: 5′-CCATGGCTTTAAGCATCGCA; Reverse Oligo: 5′-TTATGACGGATACGGTTACGG. Synthesis fragment: 5′- GGTTGTCATCGGCAATTTTGTGgtgagtactcttatcgaactgctgacttgtaaacgatgtttactggctat aatgctgacttatcgcct.

The *Drosophila* injection strain was *white*[1118]. 206 embryos were injected. 36 G0 crosses were established. Of the 78 positive lines crosses, in the F1 screen 25 positive lines were identified. Seven lines were positively validated by PCR and 1 line was sequenced confirming no unexpected changes in *para*. Lines were isogenised and balanced. DsRed was excised using PiggyBac (PBac) Transposase Bloomington Stock RRID:BDSC_8285. Excision was validated by genomic PCR and sequencing. Resulting lines were hemizygous viable. The line used in study had internal identifier: 20256ex1.

## Choice, handling, and preparation of drugs

The initial preliminary analysis was conducted using a group of 12 compounds 'proof of principle' compounds and a solvent control. These compounds were initially used to compare both the video method and ethoscope method. After testing these initial compounds, it was found that the ethoscope methodology was more successful, and then the compound list was expanded to 17 (including the control) only using the ethoscope method. As a final test, we included additional compounds for a single concentration, bringing up the total to 40 (including control), also for the ethoscope method. All insecticide compounds were supplied by Syngenta Ltd from their in-house stock (see *Supplementary file 1* for a full list of compounds used). Compounds were received in solid form and diluted in solvent containing 5% ethanol (VWR, 20821), 5% acetone (Sigma, 179124), and 10% dimethylsulfoxide (D2650, Sigma) in distilled water to 1000 ppm initially, then further diluted in the solvent mixture to 100 and 1 ppm (where 1 ml/l = 1000 ppm). For insecticide assays, 0.5 ml of 5% sucrose (Sigma, S0389), 1% agarose (Sigma, A6236) solution was pipetted into each well and allowed to set. Following this, 2 µl of compound solution were placed on the surface and allowed to dry for 30 min or more. Male flies were then placed on the surface with a small glass cover slip placed on top (13 mm circular cover slip, VWR631-0150). Flies were briefly anaesthetised (>1 min) before being placed onto the surface of the plate. Once each well had been filled with a single male fly, arenas were placed into the ethoscope and recorded for a minimum of 2 days. All experiments were started between ZT0 and ZT1 and within 30 min of the flies being placed in the wells. For each compound in *Figure 1c* and *Figure 1—figure supplement 1c*, three repeats were done at different time points. For *Figure 1d*, two repeats to three biological repeats per compound.

## Data acquisition and processing

Ethoscope data were first processed in R using rethomics (*Geissmann et al., 2019b*). Each time series was exported (.csv) and converted using Python to individual time series in a file format (.dat) compatible with MatLab. A metadata (.txt) file served as a reference file of each individual time series with keywords outlining compound groups and concentrations for processing data using HCTSA.

## HCTSA/Catch22

Following this process, HCTSA feature extraction was performed on the time-series data (for *Figure 1c, d* and *Figure 3c, d*, *Figure 1—figure supplements 1 and 2a*). After the features were extracted, outputs of error-producing operations were removed through a normalisation process using a sigmoidal transformation. HCTSA inbuilt functions were then used to classify data using a linear SVM classifier and a confusion matrix comparing the time series was generated. For some of the time-series data (that in *Figure 2b, d*, *Figure 1—figure supplement 2b*), a smaller feature set of HCTSA, Catch-22 was used for feature extraction. Due to the smaller number of features used with this method, normalisation was not required before using a linear SVM classifier to generate a confusion matrix comparing the results. A time series of 12 hr was used for the HCTSA analysis in *Figure 1* and *Figure 1—figure supplements 1 and 2*. A length of 3 hr was used for the HCTSA analysis in *Figure 3*. All video data in *Figure 2* are from time series of 15 min. By always running the full set of features on aggregate to train a classifier (e.g. TS_Classify in HCTSA), no post hoc correction is necessary because the trained classifier only ever makes a single prediction (only one test is performed).

## Video generation for flies on insecticides

Custom 3D-printed squares were designed using the online CAD software Onshape and printed using Ultimaker 2+ 3D printers using PLA plastic. For insecticide assays, 0.5 ml of 5% sucrose (Sigma, S0389), 1% agarose (Sigma, A6236) solution was pipetted into each well and allowed to set. Following this, 2 µl of compound solution were placed on the surface and allowed to dry for 30 min or more before male flies were placed on the surface with a small glass cover slip placed on top (13 mm circular cover slip, VWR631-0150). Flies were briefly anaesthetised (>1 min) before being placed onto the surface of the square. Once each well had been filled with a single male fly, squares were placed in the arena and a video was recorded for a minimum of 12 hr using an ELP 8 megapixel camera with an IMX179 Sensor and 2.8–12 mm variable focus manual lens. All video recordings were started between ZT0 and ZT1. Recordings of flies exposed to compounds were done in a randomised manner. Video data were then broken down into shorter segments of 15 min videos for processing. The first 15 min following 1 hr of fly exposure to compound or control was used for pose-extraction.

## DeepLabCut/B-SoID

The use of DeepLabCut (version 2.1) followed the detailed protocol outlined by *Nath et al., 2019*. Briefly, frames for labelling were extracted from 3 representative videos using a *K*-means algorithm and frames were labelled with 22 unique body parts (head, left eye, right eye, thorax top, thorax bottom, abdomen top, abdomen middle, abdomen bottom, left wing tip, right wing tip, left foreleg tip, left foreleg middle, right foreleg tip, right foreleg middle, left middle leg tip, left middle leg middle, right middle leg tip, right middle leg middle, left back leg tip, left back leg middle, right back leg tip, and right back leg middle). These frames were labelled locally with a DeepLabCut graphical user interface before the project file was uploaded to Google Drive for training and video analysis to be done using Google Colab. The data were split into a 9:1 test:train dataset and training was run for more than 150,000 iterations before the average Euclidean error was computed between labels and predictions. The model at the best performing checkpoint was used to predict pose in novel videos. Following this, B-SoID was used to de-structure behaviour using the output of DeepLabCut and generate fly-specific time series of behavioural grammar as in *Hsu and Yttri, 2021*.

## Sleep deprivation

Maximal velocity time-series data generated from recording flies exposed to either control conditions (no SD) or incrementally increasing immobility-triggered SD conditions were taken from the dataset generated by *Geissmann et al., 2019a* and analysed as above. Only data for female flies were included in this study, limiting to a sample of 60 individuals per experimental group. Wherever experimental groups consisted of more than 60 individuals, 60 individual flies were randomly chosen.

## Survival

Flies were fed with the specified compounds at the desired concentrations and concomitantly analysed in ethoscopes for 24 hr. Time of death was calculated as the last moment of detected motion.

## Acknowledgements

We thank the Gilestro lab at Imperial College London and Robert Lind at Syngenta for useful discussions. Special thanks to Laurence Blackhurst for compiling the catch22 notebooks. HJ was supported by a BBSRC/CASE studentship in partnership with Syngenta (project reference BB/M011178/1/1958700). Stocks obtained from the Bloomington *Drosophila* Stock Center (NIH P40OD018537) were used in this study.

## Additional information

### Funding

| Funder | Grant reference number | Author |
| --- | --- | --- |
| Biotechnology and Biological Sciences Research Council | BB/M011178/1 | Hannah Jones |

The funders had no role in study design, data collection, and interpretation, or the decision to submit the work for publication.

### Author contributions

Hannah Jones, Data curation, Investigation, Methodology, Project administration; Jenny A Willis, Lucy C Firth, Resources, Validation; Carlo NG Giachello, Methodology; Giorgio F Gilestro, Conceptualization, Formal analysis, Supervision, Funding acquisition, Investigation, Methodology, Writing – original draft, Project administration, Writing – review and editing

### Author ORCIDs

Hannah Jones http://orcid.org/0000-0002-9481-8094
Carlo NG Giachello https://orcid.org/0000-0003-4186-1571
Giorgio F Gilestro https://orcid.org/0000-0001-7512-8541

Joint Public Review: https://doi.org/10.7554/eLife.86695.3.sa1
Author Response https://doi.org/10.7554/eLife.86695.3.sa2

## Additional files

### Supplementary files

• Supplementary file 1. Table listing of all the compounds used in this study, each with its relative bibliographic reference.

• Supplementary file 2. Two Jupyter notebooks guiding the user through the integration of ethoscope data with highly comparative time-series analysis (HCTSA; notebook 1) and Catch22 (notebook 2).

• MDAR checklist

### Data availability

A notebook version of the source code used to generate all figures is available on the Zenodo public repository, along with all the metadata and the raw data collected in this study (DOIs: 10.5281/zenodo.7335575 and 10.5281/zenodo.7393689). Data were analysed using rethomics (*Geissmann et al., 2019b*) and ethoscopy (*Blackhurst et al., 2023*). Two notebooks showing how to use the system for multiple uses are provided in *Supplementary file 2*.

The following datasets were generated:

| Author(s) | Year | Dataset title | Dataset URL | Database and Identifier |
|---|---|---|---|---|
| Jones H, Gilestro G | 2022 | A reductionist paradigm for high-throughput behavioural fingerprinting in *Drosophila melanogaster* - DATASET 1 of 2 | https://doi.org/10.5281/zenodo.7335575 | Zenodo, 10.5281/zenodo.7335575 |
| Jones H, Gilestro G | 2022 | A reductionist paradigm for high-throughput behavioural fingerprinting in *Drosophila melanogaster* - DATASET 2 of 2 | https://doi.org/10.5281/zenodo.7393689 | Zenodo, 10.5281/zenodo.7393689 |

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
