## [Editor Report · eLife assessment]

This study presents an **important** open-source resource for high-throughput behavioral screening. The protocols employ inexpensive, off the shelf hardware, and allow real-time analysis of hundreds of behaving flies. Although these protocols were developed using *Drosophila melanogaster*, they could easily be applied to other models. The evidence in support of the conclusions is **solid** and the revisions carried out by the authors go a long way towards providing the user with an integrated system that is also more user-friendly.

---

## [Referee Report · Joint Public Review]

In the current paper, Jones et al. describe a new framework, named "coccinella", for real-time high-throughput behavioral analysis aimed at reducing the cost of analyzing behavior. In the setup used here each fly is confined to a small circular arena and able to walk around on an agar bed spiked with nutrients or pharmacological agents. The new framework, built on the researchers' previously developed platform Ethoscope, relies on relatively low-cost Raspberry Pi video cameras to acquire images at ~0.5 Hz and pull out, in real time, the maximal velocity (parameter extraction) during 10 second windows from each video. Thus, the program produces a text file, and not voluminous videos requiring storage facilities for large amounts of video data, a prohibitive step in many behavioral analyses. The maximal velocity time-series is then fed to an algorithm called Highly Comparative Time-Series Classification (HCTSA)(which itself is based on a large number of feature extraction algorithms) developed by other researchers. HCTSA identifies statistically salient features in the time-series which are then passed on to a type of linear classifier algorithm called support vector machines (SVM). In cases where such analyses are sufficient for characterizing the behaviors of interest this system performs as well as other state-of-the-art systems used in behavioral analysis (e.g., DeepLabCut)

In a pharmacobehavior paradigm testing different chemicals, the authors show that coccinella can identify specific compounds as effectively as other more time-consuming and resource-consuming systems.

The new paradigm should be of interest to researchers involved in drug screens, and more generally, in high-throughput analysis focused on gross locomotor defects in fruit flies such as identification of sleep phenotypes. By extracting/saving only the maximal velocity from video clips, the method is fast. However, the rapidity of the platform comes at a cost--loss of information on subtle but important behavioral alterations. When seeking subtle modifications in animal behavior, solutions like DeepLabCut, which are admittedly slower but far superior in terms of the level of details they yield, would be more appropriate.

The manuscript reads well, and it is scientifically solid. The comments listed below were directed to the original submission and were satisfactorily addressed in the revised version.

1- The fact that Coccinella runs on Ethoscopes, an open source hardware platform described by the same group, is very useful because the relevant publication describes Ethoscope in detail. However, the current version of the paper does not offer details or alternatives for users that would like to test the framework, but do not have an Ethoscope. Would it be possible to overcome this barrier and have coccinella run with any video data (and, thus, potentially be used to analyze data obtained from other animal models)?

2- Readers who want background on the analytical approaches that the platform relies on following maximal velocity extraction, will have to consult the original publications. In particular, the current manuscript does not provide much explanation on Highly Comparative Time-Series Classification (HCTSA) or SVM; this may be reasonable because the methods were developed earlier by others. While some readers may find that the lack of details increases the manuscript's readability, others may be left wanting to see more discussion on these not-so-trivial approaches. In addition, it is worth noting that the same authors that published the HCTSA method, also described a shorter version named catch22, that runs faster with a similar output. Thus, explaining in more detail how HCTSA operates, considering is a relatively new method, will make the method more convincing.

---

## [Author Response]

The following is the authors’ response to the original reviews.

We thank the editor and the reviewers for their very useful and constructive comments. We went through the list and gladly received all their suggestions. The reviewers mostly pointed to minor revisions in the text, and we acted on all of those. The one suggestion that required major work was the one raised in point 13, about the processing pipeline being unconvincingly scattered between different tools (R → Python → Matlab). I agree that this was a major annoyance, and I am happy to say we have solved it integrating everything in a recent version of the ethoscopy software (available on biorxiv with DOI https://www.biorxiv.org/content/10.1101/2022.11.28.517675v2 and in press with Bioinformatics Advances). End users will now be able to perform coccinella analysis using ethoscopy only, thus relying on nothing else but Python as their data analysis tool. This revised version of the manuscript now includes two Jupyter Notebooks as supplementary material with a “pre-cooked” sample recipe of how to do that. This should really simplify adoption and provides more details on the pipeline used for phenotyping.

Please find below a point-by-point description of how we incorporated all the reviewers’ excellent suggestions.

Recommendations for the authors: please note that you control which, if any, revisions, to undertake

1. Line 38: "collecting data simultaneously from a large number of individuals with no or limited human intervention" is a bit misleading, as the entire condition the individuals are put in are highly modified by humans and most times "unnatural". I understand the point that once the animals are placed in these environments, then recording takes place without intervention, but it would be nice to rephrase this so that it reflects more accurately what is happening.

We have now rephrased this into the following (L39):

Collecting data simultaneously from a large number of individuals, which can remain undisturbed throughout recording.

1. Line 63: please add a reference to the Ethoscopes so that readers can easily find it.

Done.

(2b) And also add how much they cost and the time needed to build them, as this will allow readers to better compare the proposed system against other commercially available ones.

This information is available on the ethoscope manual website (http://lab.gilest.ro/ethoscope). The price of one ethoscope, provided all necessary tools are available, is around ~£75 and the building time very much depends on the skillset of the builder and whether they are building their first ethoscope or subsequent ones. In our experience, building and adopting ethoscopes for the first time is not any more time-expensive than building a (e.g.) deeplabcut setup for the first time. We have added this information to L81

Ethoscopes are open source and can be manufactured by a skilled end-user at a cost of about £75 per machine, mostly building on two off-the-shelf component: a Raspberry Pi microcomputer and a Raspberry Pi NoIR camera overlooking a bespoke 3D printed arena hosting freely moving flies.

1. Line 88: The authors describe that in the current setting, their system is capable of an acquisition rate of 2.2 frames per second (FPS). Would reducing the resolution of the PiCamera allow for higher FPS? I raise this point because the authors state that max velocity over a ten second window is a good feature for classifying behaviors. However, if animals move much faster than the current acquisition rate, they could, for instance, be in position X, move about and be close to the initial position when the next data point is acquired, leading to a measured low max velocity, when in fact the opposite happened. I think it would be good to add a statement addressing this (either data from the literature showing that the low FPS does not compromise data acquisition, or a test where increasing greatly FPS leads to the same results).

We have previously performed a comparison of data analysed using videos captured at different FPSs, which is published in Quentin Geissman’s doctoral Thesis (2018, DOI: https://doi.org/10.25560/69514 in chapter 2, section 2.8.3, figure 2.9 ). We have now added this work as one of the references at L95 (reference 19).

1. Still on the low FPS, would a Raspberry Pi 4 help with the sampling rate? Given that they are more powerful than the RPi3 used in the paper?

It would, but it would be a minor increase, leading from 2.2 to probably 3-5 FPS. A significantly higher number of FPSs would be best achieved by lowering the camera’s resolution, as the reviewer’s suggested, or by operating offline. I think the interesting point being implied by the reviewers is that, for *Drosophila*, the current limits of resolution are more than sufficient. For other animals, perhaps moving more abruptly, they may not. The reviewer is right that we should add a line of caveat about this. We now do so in the discussion, lines 215-224.

Coccinella is a reductionist tool, not meant to replace the behavioural categorization that other tools can offer but to complement it. It relies on raspberry PIs as main acquisition devices, with associated advantages and limitations. Ethoscopes are inexpensive and versatile but have limitations in terms of computing power and acquisition rates. Their online acquisition speed is fast enough to successfully capture the motor activity of different species of *Drosophilae28*, but may not be sufficient for other animals moving more swiftly, such as zebrafish larvae. Moreover, coccinella cannot apply labels to behaviour (“courting”, “lounging”, “sipping”, “jumping” etc.) but it can successfully identify large behavioural phenotypes and generate unbiased hypothesis on how behaviour – and a nervous system at large – can be influenced by chemicals, genetics, artificial manipulations in general.

1. Along the same line of thought, would using a simple webcam (with similar specs to the PiCamera - ELP has cameras that operate on infrared and are quite affordable too) connected to a more powerful computer lead to higher FPS? - The reason for the question about using a simple webcam is that this would make your system more flexible (especially useful in the current shortage of RPi boards on the market) lowering the barrier for others to use it, increasing the chances for adoption.

Completely bypassing ethoscopes would require the users to setup their own tracking solution, with a final result that may or may not match what we describe here. If a greater temporal resolution is necessary, the easiest way to achieve more FPSs would be to either decrease camera resolution or use the Pis to take videos offline and then process those videos at a later stage. The combination of these two would give FPS acquisition of 60 fps at 720p, which is the maximum the camera can achieve. We now made this clear at lines 83-92.

The temporal and spatial resolution of the collected images depends on the working modality the user chooses. When operating in offline mode, ethoscopes are capable to acquire 720p videos at 60 fps, which is a convenient option with fast moving animals. In this study, we instead opted for the default ethoscope working settings, providing online tracking and realtime parametric extraction, meaning that images are analysed by each raspberry Pi at the very moment they were acquired (Figure 1b). This latter modality limits the temporal resolution of information being processed (one frame every 444 ms ± 127 ms, equivalent to 2.2 fps on a Raspberry Pi3 at a resolution of 1280x960 pixels with each animal being constricted in an ellipse measuring 25.8 ± 1.4 x 9.85 ±1.4 pixels - Figure 1a) but provides the most affordable and high-throughput solution, dispensing the researcher from organising video storage or asynchronous video processing for animals tracking.

1. One last point about decreasing use barrier and increasing adoption: Would it be possible to use DeepLabCut (DLC) to simply annotate each animal (instead of each body part) and feed the extracted data into your current analysis with coccinella? This way different labs that already have pipelines in place that use DLC would have a much easier time in testing and eventually switching to coccinella? I understand that extracting simple maximal velocity this way would be an overkill, but the trade-off would again be a lowering of the adoption barrier.

It would certainly be possible to calculate velocity from the whole animal pose measurement and then use this with HCTSA or Catch22, thus mimicking the coccinella pipeline, but it would be definitely overkilled, as the reviewers correctly points out. Given that we are trying to make an argument about high-throughput data acquisition I would rather not suggest this option in the manuscript.

1. Line 96: The authors state that once data is collected, it is put through a computational frameworkthat uses 7700 tests described in the literature so that meaningful discriminative features are found. I think it would be interesting to expand a bit on the explanation of how this framework deals multiple comparison/multiple testing issues.

We always use the full set of features on aggregate to train a classifier (e.g., TS_Classify in HCTSA) and that means no correction is necessary because the trained classifier only ever makes a single prediction (only one test is performed), so as long as it is done correctly (e.g., proper separation of training and test sets, etc.) then multiple hypothesis correction is not appropriate. This has been confirmed with the HCTSA/Catch22 author (Dr Ben Fulcher, personal communication). We have added a clarifying sentence about this to the methods (L315-318)

1. It would be nice to have a couple of lines explaining the choice of compounds used for testing and also why in some tests, 17 compounds were used, while in others 40, and then 12? I understand how much work it must be in terms of experiment preparation and data collection for these many flies and compounds, but these changes in the compounds used for testing without a more detailed explanation is suboptimal.

This is another good point. We have now added this information to the methods, in a section renamed “choice, handling and preparation of drugs” L280-285, which now reads like this:

The initial preliminary analysis was conducted using a group of 12 compounds “proof of principle” compounds and a solvent control. These compounds were initially used to compare both the video method and ethoscope method. After testing these initial compounds, it was found that the ethoscope methodology was more successful, and then the compound list was expanded to 17 (including the control) only using the ethoscope method. As a final test, we included additional compounds for a single concentration, bringing up the total to 40 (including control), also for the ethoscope method.

1. Line 119 states: "A similar drop in accuracy was observed using a smaller panel of 12 treatments (Supplementary Figure 2a)". It is actually Supplementary Figure 1c.

Thank you for noticing that! Now corrected. The Supplementary figures have also been renamed to obey eLife’s expected nomenclature (both Figure 1 – Figure supplements)

1. In some places the language seems a little outlandish and should either be removed or appropriately qualified. a- Lines 56-59 pose three questions that are either rhetorical or ill-posed. For example, "...minimal amount of information...behavior" implies there is a singular response but the response depends on many details such as to what degree do the authors want to "classify behavior".

Yes, those were meant as rhetorical questions indeed, but we prefer to keep them in, because we are hoping to generate this type of thoughts with the readers. These are concepts that may not be so obvious to someone who is just looking to apply an existing tool and may spring some reflection about what kind of data do they really want/need to acquire.

b) Some of the criticisms leveled at the state-of-the-art methods are probably unwarranted because the goals of the different approaches are different. The current method does not yield the type of rich information that DeepLabCut yields. So, depending on the application DeepLabCut may be the method of choice. The authors of the current manuscript should more clearly state that.

In the introduction and discussion we do try to stress that coccinella is not meant to replace tools like DLC. We have now added more emphasis to this concept, for instance to L212:

[tools like deeplabcut] are ideal – and irreplaceable – to identify behavioural patterns and study fine motor control but may be undue for many other uses.

And L215:

Coccinella is a reductionist tool not meant to replace the behavioural categorization that other tools can offer but to complement it

1. The application to sleep data appears suddenly in the manuscript. The authors should attempt to make with text change a smoother transition from drug screen to investigation into sleep.

I agree with this observation. We have now tried to add a couple of sentences to contextualise this experiment and hopefully make the connection appear more natural. Ultimately, this is a proof-ofprinciple example anyway so hopefully the reader will take it for what it is (L169).

Finally, to push the system to its limit, we asked coccinella to find qualitative differences not in pharmacologically induced changes in activity, but in a type of spontaneous behaviour mostly characterised by lack of movement: sleep. In particular, we wondered whether coccinella could provide biological insights comparing conditions of sleep rebound observed after different regimes of sleep deprivation. *Drosophila melanogaster* is known to show a strong, conserved homeostatic regulation of sleep that forces flies to recover at least in part lost sleep, for instance after a night of forceful sleep deprivation.

(11b) Additionally, the beginning section of sleep experiments talks about sleep depth yet the conclusion drawn from sleep rebound says more about the validity of the current 5 min definition of sleep than about sleep depth. If this conclusion was misunderstood, it should be clarified. If it was not, the beginning text of the sleep section should be tailored to better fit the conclusion.

I am afraid we did not a good job at explaining a critical aspect here: the data fed to coccinella are the “raw” activity data, in which we are not making any assumption on the state of the animal. In other words, we do not use the 5-minutes at this or any other point to classify sleep and wakening. Nevertheless, coccinella picks the 300 seconds threshold as the critical one for discerning the two groups. This is interesting because it provides a full agnostic confirmation of the five minutes rule in *D. melanogaster*. We recognise this was not necessarily obvious from the text and now added a clarification at L189-201:

However, analysis of those same animals during rebound after sleep deprivation showed a clear clustering, segregating the samples in two subsets with separation around the 300 seconds inactivity trigger (Figure 3d). This result is important for two reasons: on one hand, it provides, for the third time, strong evidence that the system is not simply overfitting data of nought biological significance, given that it could not perform any better than a random classifier on the baseline control. On the other hand, coccinella could find biologically relevant differences on rebound data after different regimes of sleep deprivation. Interestingly enough, the 300 seconds threshold that coccinella independently identified has a deep intrinsic significance for the field, for it is considered to be the threshold beyond which flies lose arousal response to external stimuli, defining a “sleep quantum” (i.e.: the minimum amount of time required for transforming inactivity bouts into sleep bouts23,24,28). Coccinella’s analysis ran agnostic of the arbitrary 5-minutes threshold and yet identified the same value as the one able to segregate the two clusters, thus providing an independent confirmation of the fiveminutes rule in *D. melanogaster*.

1. Line 227: (standard food) - please add a link to a protocol or a detailed description on what is "standard food". This way others can precisely replicate what you are using. This is not my field, but I have the impression that food content/composition for these animals makes big changes in behaviour?

Yes, good point. We have now added the actual recipe to the methods L240:

Fly lines were maintained on a 12-hour light: 12-hour dark (LD) cycle and raised on polenta and yeast-based fly media (agar 96 g, polenta 240 g, fructose 960 g and Brewer’s yeast 1,200 g in 12 litres of water).

1. Data acquisition and processing: please add links to the code used.

Both the code and the raw data used to generate all the figures have been uploaded on Zenodo and available through their repository. Zenodo has a limit of 50GB per uploaded dataset so we had to split everything into two files, with two DOIs, given in the methods (L356, section “code and availability” - DOIs: 10.5281/zenodo.7335575 and 10.5281/zenodo.7393689). We have now also created a landing page for the entire project at http://lab.gilest.ro/coccinella and linked that landing page in the introduction (L64).

13b) Also your pipeline seems to use three different programming languages/environments... Any chance this could be reduced? Maybe there are R packages that can convert csv to matlab compatible formats, so you can avoid the Python step? (nothing against using the current pipeline per se, I am just thinking that for usability and adoption by other labs, the smaller amount of languages, the better?

This is a very important suggestion that highlights a clear limitation of the pipeline. I am happy to say that we worked on this and solved the problem integrating the Python version of Catch22 into the ethoscopy software. This means the two now integrate, and the entire analysis can be run within the Python ecosystem. HCTSA does not have a Python package unfortunately but we still streamlined the process so that one only has to go from Python to Matlab without passing through R. To be honest, Catch22 is the evolution of HCTSA and performs really well so I think that is what most users will want to use. We provide two supplementary notebooks to guide the reader through the process. One explains how to go from ethoscope data to an HCTSA compatible mat file. The other explains how ethoscope data integrate with Catch22 and provides many more examples than the ones found in the paper figures.

1. There are two sections named "References" (which are different from each other) on the manuscript I received and also on BioRxiv. Should one of them be a supplementary reference? Please correct it. I spent a bit of time trying to figure out why cited references in the paper had nothing to do with what was being described...

The second list of references actually applied only to the list of compounds in the supplementary table 1. When generating a collated PDF this appeared at the end of the document and created confusion. We have now amended the heading of that list in the following way, to read more appropriately: